# The Etiology of Cholelithiasis in Children and Adolescents—A Literature Review

**DOI:** 10.3390/ijms232113376

**Published:** 2022-11-02

**Authors:** Katarzyna Zdanowicz, Jaroslaw Daniluk, Dariusz Marek Lebensztejn, Urszula Daniluk

**Affiliations:** 1Department of Pediatrics, Gastroenterology, Hepatology, Nutrition and Allergology, Medical University of Bialystok, 15-274 Bialystok, Poland; 2Department of Gastroenterology and Internal Medicine, Medical University of Bialystok, 15-276 Bialystok, Poland

**Keywords:** gallstones, cholelithiasis, children, pathogenesis

## Abstract

The incidence of gallstone disease has increased in recent years. The pathogenesis of cholelithiasis is not fully understood. The occurrence of the disease is influenced by both genetic and environmental factors. This article reviews the literature on cholelithiasis in children, with the exception of articles on hematological causes of cholelithiasis and cholelithiasis surgery. The aim of this review is to present the latest research on the pathogenesis of gallstone disease in children. The paper discusses the influence of all factors known so far, such as genetic predisposition, age, infections, medications used, parenteral nutrition, and comorbidities, on the development of gallstone disease. The course of cholelithiasis in the pediatric population is complex, ranging from asymptomatic to life-threatening. Understanding the course of the disease and predisposing factors can result in a faster diagnosis of the disease and administration of appropriate treatment.

## 1. Introduction

The broad spectrum of biliary tract disease includes the most often diagnosed gallstone disease (cholelithiasis), cholecystitis, or biliary dyskinesia [1,2]. Gallstones are formed mainly in the gallbladder, less often in intrahepatic or extrahepatic bile ducts. More patients with cholelithiasis have no obvious symptoms. Symptomatic patients presented with dyspepsia and biliary colic caused mainly by obstruction of the cystic duct [3]. Gallstones may lead to serious complications such as cholecystitis, acute cholangitis, and pancreatitis [1]. In recent years, the prevalence of cholelithiasis has risen and ranges from 1.9% to 4% in children [4,5]. This increase may be caused by the worrying problem of childhood obesity and the widespread use of ultrasound [6,7,8]. Simultaneously, in pediatric patients, the number of performed cholecystectomies rose by 213% over a 9-year period [9]. Additionally, cases of fetal cholelithiasis were also described [10,11]. Schwab et al., in approximately 200,000 obstetric sonograms taken between 1996 and 2019, found 34 fetal cholelithiases. The median gestational age at diagnosis was 34.7 weeks, and the majority of cases were female (56%) [10]. Troyano-Luque et al. reported two cases of fetal cholelithiasis. In one of them, cholelithiasis was diagnosed 6 weeks after 16 days of ceftriaxone therapy in maternal Lyme disease [11]. The presence of gallstones in fetuses may lead researchers to look for genetic causes of the disease.

Based on their composition, gallstones are classified into pigment stones, cholesterol stones, and mixed stones. Pigment stones are mainly observed in hemolytic diseases, and their incidence remains stable [9]. Cholesterol gallstones are caused by genetic and environmental factors leading to an elevated concertation of cholesterol in the bile.

Cholesterol plays a structural and metabolic role. It is a component of membrane cells, steroid hormones, and a bile acid precursor. The level of cholesterol in the human body depends on endogenous de novo synthesis, enterohepatic recirculation, and dietary intake [12]. The liver is an organ that is significantly involved in cholesterol metabolism through its synthesis mediated by 3-hydroxy-3-methylglutaryl coenzyme A reductase (HMGR), cholesterol uptake by receptors (low-density lipoprotein receptors (LDLr), the prolow-density lipoprotein receptor-related protein 1 (LRP1), the scavenger receptor class B member 1 (SRB1), release of lipoproteins into the bloodstream, conversion of cholesterol into bile acids or cholesterol excretion into bile. Cholesterol is secreted by adenosine triphosphate protein binding cassette transporters G5/G8 (ABC G5/G8) to the biliary tract [13]. As a hydrophilic substance, it is solubilized in mixed micelles and vesicles that consist of bile acids and phosphatidylcholine [14].

Intestinal absorption is relevant for maintaining cholesterol balance. Cholesterol is absorbed by the Niemann–Pick C1-like 1 (NPC1L1) transporter, which, together with ABCG5 / G8, is located in the enterocyte brush border membrane. Intracellular cholesterol is esterified by the acetyl-CoA cholesterol acyltransferasen 2 (ACAT-2) and incorporated along with triglycerides, phospholipids, and apolipoprotein B-48 into chylomicrons. Next, through the lymphatic system and bloodstream, the chylomicron remnants may be absorbed by the liver. Additionally, ABCG5/G8 proteins promote the active efflux of non-esterified cholesterol and plant sterols from the enterocytes into the intestinal lumen for fecal excretion [15].

In hepatocytes, cholesterol is covered in bile acids through classical and alternative pathways. The first is the dominant one, mediated by 7α-hydroxylase (CYP21A1). The alternative pathway is controlled by sterol 27-hydroxylase (CYP27A1) and is responsible for the production of 9% to 25% of bile acids. Chenodeoxycholic acid has produced both ways, and cholic acid is only synthesized in a classic way. Newly produced bile acids, in conjunction with glycine and taurine, are pumped out by the ATP-binding cassette, sub-family B member 11 (ABC B11) [16].

Bile-binding acids are reabsorbed by the apical sodium-dependent bile acid transporter (ASBT) in the terminal ileum and then secreted into the portal vein by the organic solute transporter α/β (OST α/β). In the liver, conjugated bile acids return to the liver via Na+/taurocholic acid cotransport polypeptide (NTCP). Members of the Organic Anion Transporting Polypeptide (OATP) family are responsible for the uptake of unconjugated or sulfated bile acids. Enterohepatic circulation provides approximately 95% of bile acid return [16,17].

In the human body, cholesterol synthesis equals its secretion as bile acid. However, it may be disturbed by food intake and diseases [17]. The state of homeostasis is regulated by the expression of genes responsible for cholesterol transport and metabolism, transcription factors, and posttranscriptional regulatory circuits. The farnesoid X receptor (FXR) and the liver X receptors (LXRs) are nuclear receptors that play an important role in the regulation of genes encoding ABC transporters [18]. Lowering cellular cholesterol activates the endoplasmic reticulum membrane-bound transcription factor, sterol regulatory element-binding protein isoform 2 (SREBP-2), leading to enhanced cholesterol uptake and biosynthesis through regulation of genes encoding low-density lipoprotein (LDL) receptor and HMGR. On the contrary, an increase in the level of cellular cholesterol promotes (LXRs) transcription factors to enhanced cholesterol efflux from the liver [19]. The expression of CYP7A1 in the liver is regulated by the fibroblast growth factor 19 (FGF19). FGF19 is activated by bile acid in the ileum mediated by the action of ASBT. The inhibition of ASBT causes an increase in FGF19 levels and the synthesis of bile acids [12].

The risk factors for the development of cholelithiasis include age, gender, body weight, comorbidities, diet, and physical activity [20]. Although these factors are recognized in the adult population, they are modified in children. The formation of cholesterol gallstones is triggered by excessive cholesterol concentration in bile. Defects responsible for this condition are genes, hypersecretion of liver cholesterol, rapid phase transitions of cholesterol in bile, dysmotility of the gallbladder, and intestinal factors [20].

The aim of the review is to present the latest research on the pathogenesis of cholelithiasis in children. To date, many studies on the development and incidence of gallstone disease in adults have been published. To the best of our knowledge, there are no reviews of the pathomechanism of cholelithiasis development in children. A review of MEDLINE/PubMed data was carried out in August 2022, using the phrase ‘cholelithiasis’ or ‘gallstones’ in combination with ’children,’ ‘adolescents,’ or ‘pediatrics.’ Screening of the titles and abstracts was independently performed by two investigators. The studies included in this analysis fulfilled the inclusion criteria: they explored the etiology of cholelithiasis in pediatric patients and had available full texts. We excluded from the analysis articles on hematological causes of gallstone disease and surgical procedures for cholelithiasis. The selected papers were discussed with all authors.

## 2. Review

### 2.1. Genes

A Swedish twin study showed that an inherited predisposition is responsible for 25% of the overall risk of developing gallstones [21]. Lithogenic genes 1 and 2 (*Lith1* and *Lith2*), playing a role in liver cholesterol secretion and regulating bile flow, have been described in murine models. Their human counterparts are *ABCG5* and *ABCG8* [17]. A recent study of 214 children with cholelithiasis showed the presence of the lithogenic *ABCG8* allele p.D19H in 14.9% of children, which was more frequently reported compared to children and adults without gallstones. Additionally, carriers of one copy of the lithogenic variant p.D19H were also at higher risk of the development of gallstones. Increased susceptibility to the formation of cholesterol stones is associated with abnormal cholesterol metabolism resulting from its increased transport or lower intestinal absorption in combination with increased cholesterol synthesis [22]. MicroRNAs (miRNAs) are small noncoding RNAs that regulate gene expression. Recent studies have shown that miRNA-223, can prevent the development of gallstones in mice on a lithogenic diet by directly affecting the ABCG5 and ABCG8 transporters. To date, no studies have been published assessing the effect of miRNA-223 on the development of gallstone disease in humans [23].

Krawczyk et al. also found that *NPC1L1* rs217434 polymorphism also was connected with the occurrence of gallstones (only in comparison with healthy adults) and lower campesterol: desmosterol ratio. However, the *UGT1A1* genotype did not differ between children with and without cholelithiasis [22].

Nissinen et al. determined the D19H polymorphism of the *ABCG8* gene, serum cholesterol, non-cholesterol sterols, and lipids in 66 children affected by gallstones in later life and 126 children from the control group. In the first group, 22.7% of patients carried the *ABCG8* 19H allele, and in the control group, this percentage was 19.0%. A decrease in phytosterols was observed in patients with the lithogenic variant. According to this study, low phytosterols in childhood promoted the occurrence of cholelithiasis in adults in carriers of the risk variant 19H of the *ABCG8* gene. What is more, *NPC1L1* variants: -18C > A (rs41279633) and V1296V T > C (rs2174340) had a minor influence on non-cholesterol sterols [24].

In another genetic study, the *ABCB4* gene (encoding a multi-drug resistance protein 3 (MDR3) was evaluated in the pathogenesis of idiopathic gallstones. The mutation of the *ABCB4* may lead to low-phospholipid associated cholelithiasis (LPAC) defined by the presence of symptomatic and recurrent cholelithiasis in young patients with abnormal ultrasound of the liver, progressive familial intrahepatic cholestasis (PFIC) type 3, low phospholipid-associated cholelithiasis, and intrahepatic cholestasis of pregnancy [25]. In a retrospective analysis of 26 pediatric patients with genetically proven mutations of the *ABCB4* gene, gallstone disease was diagnosed in 15% of patients, but in adults, this level was higher (67% of patients) [26]. In another study involving 35 children with idiopathic gallstones meeting the clinical criteria of LPAC, in only one case, a possibly pathogenic variant c.2318G > T of the *ABCB4* gene was found. This phenomenon may be explained by sexual immaturity, which may affect the course of LPAC [27]. In an analysis by Krawczyk et al., *ABCB4* c.504C > T and *ABCB4* c.711A > T alleles were not associated with the development of cholelithiasis [22].

The etiology of gallstones may include NTCP deficiency, which is encoded by the *SLC10A1* gene. Dong et al., in a group of 13 children with mutations of the *SLC10A1* gene, in one case (male, 10 months old) cholelithiasis was diagnosed [28]. Similarly, Mao et al. described in two infants with *NTCP* p.Ser267Phe variant the presence of cholelithiasis. Interestingly, NTCP knockout (*SLC10A1^−/−^*) mice develop multiple abnormal phenotypes of the gallbladder and hypercholanemia, but not cholelithiasis [29]. This data may suggest that abnormalities in bile acid metabolism caused by NTCP deficiency predispose to gallstone formation.

In a study of cryptogenic cholelithiasis, 17% of cases had juvenile cholelithiasis. Further genetic analyses revealed in two young adult’s pathogenic mutations in the *ATP8B1* and *ABCB11* genes. However, multivariate analysis did not show that cholelithiasis was an independent factor associated with cholestasis-causing mutations [30].

In pediatric patients with cholesterol and pigment gallstones, RNA expression of ABCG5 and ABCG8 measured by quantitative real-time reverse transcription polymerase chain reaction (qRT-PCR) showed increased levels in patients with cholelithiasis than in healthy controls. This increase was similar in patients’ cholesterol and pigment stones. However, RNA expression of FXR, ATP-binding cassette C2 (ABCC2), and ABCB4 transporters did not differ significantly between the study and control groups. Moreover, patients with cholesterol stones had decreased plant sterols (campesterol and sitosterol), and increased cholesterol precursors compared not only to healthy controls but also to children with pigment stones. Both findings explain the higher content in cholesterol cholelithiasis [31]. Similarly, in another study, plant sterols, markers of cholesterol absorption, were lower than in patients with black pigment stones [32]. Comparatively low levels of plant sterols and cholestanol have been observed in children who developed gallstones in adulthood. This observation can be used as prognostic marker of the development of gallstone disease [33].

The genetic factors that have been studied in pediatric patients with cholelithiasis are listed in Table 1.

### 2.2. Proteins and Lipids

There is evidence of the involvement of adipokines and hepatokines in the development of cholelithiasis in children. Higher levels of chemerin, retinol-binding protein 4 (RBP-4), and fibroblast growth factor 21 (FGF-21) have been observed in children with cholelithiasis. Taking into consideration the influence of adipose tissue in lean children, only chemerin was significantly increased in patients with cholelithiasis [34]. Based on the literature, chemerin may be both a pro-inflammatory and anti-inflammatory molecule [35]. The influence of other markers of inflammation was observed by Denisova et al. [36]. In this study, increased expression of various interleukins (IL-1, IL-4, IL-6, IL-7, IL-8, and IL-17A) was observed in calculous cholecystitis.

The first study on serum lipid profile in patients with gallstone disease was recently published. The authors showed that the serum values of total cholesterol (TC), sphinganine (SPA), ceramides: C14:0-Cer, C16:0-Cer, C18:1-Cer, C18:0-Cer, C20:0-Cer, C24:1-Cer and lactosylceramides: C16:0-LacCer, C18:0-LacCer, C18:1-LacCer, C24:0-LacCer, C24:1-LacCer differed significantly between patients with and without gallstones. In the generalized multivariate linear model, after taking into account age, sex, obesity, TC, and TG levels, the best differentiating sphingolipids for gallstone disease were the reduced values of SPA, C14:0-Cer, C16:0-Cer, C24:1-LacCer, C24:0-LacCer and elevated concentrations of C20:0-Cer, C24:1-Cer, C16:0-LacCer and C18:1-LacCer [37]. These results suggest that serum sphingolipids may be potential biomarkers in patients with gallstone disease.

### 2.3. Microbiome

The microbiome of the human digestive tract plays a relevant role in human health, such as nutrition and metabolism functions, preventing the invasion of infectious agents or enhancing intestinal integrity [38]. In the adult study, increment of the intestinal bacterial phylum Proteobacteria and decrement of *Faecalibacterium* spp., *Lachnospira* spp., and *Roseburia* spp. were observed in gallstone patients [39]. We did not find similar studies in the pediatric population. However, infection factors may be associated with the development of cholelithiasis. In a study of adults and children, the overall adjusted odds ratio of gallbladder stones for patients with cholelithiasis and *Clonorchis sinensis* infection compared to non-infected individuals was 2.2 (95% CI: 0.9–5.6) [40]. Another study suggests a potential connection between calcium carbonate gallbladder stones and *C. sinensis* infection [31].

### 2.4. Diet, Drugs, and Toxins

Food intake has been indicated as a potential risk factor for cholelithiasis. To the best of our knowledge, no studies describing the influence of diet on the occurrence of gallstone disease in children have been published so far. However, considering the effect of diet on the incidence of cholelithiasis in adults may be of similar importance, especially in Western countries. The increased fat intake with highly refined sugars, fructose, and low fiber contents predisposes to the development of gallstones. These eating habits lead to increased biliary cholesterol concentrations and hypertriglyceridemia-induced secretion of gallbladder mucin [41]. It is important to promote the principles of a balanced diet among children and adolescents. It seems likely that avoiding fast food and sugary drinks could help reduce the incidence of gallstone disease in children.

Parenteral nutrition is often associated with intestinal failure–associated liver disease (IFALD), including elevated markers of cholestasis, hepatic fibrosis, biliary cirrhosis, portal hypertension, and cholelithiasis. The pathogenesis of IFALD is complex and depends on components of parenteral nutrition solution and active factors absorbed from the intestine [42]. In pediatric studies, total parenteral nutrition was a risk factor in 10–17.6% of cholelithiasis cases [43,44]. According to Pichler et al., younger age at the beginning of parenteral nutrition and the primary diagnosis of motility disorder with ileostomy fashioned were predictors for gallstones. On the other hand, the lower incidence of gallstones was associated with modifications of lipid emulsions (containing less soya and added olive and/or medium-chain triglyceride (MCT) and fish oil when compared with the use of pure soya lipid) [45]. The total resolution of gallstones in one patient and a decrease in the size of gallstones in the other patient were observed during using SMOF lipid (a complex mixed emulsion of 20% lipid containing 30% soybean oil, 30% MCT, 25% monounsaturated fatty acids, and 15% fish oil) [46]. The lithogenic role of a ketogenic diet, which is a high-fat, low-carbohydrate, and moderate-protein diet, has been described [47].

Moreover, abnormal drug concentration in the biliary tract may promote the development of cholelithiasis. Ceftriaxone, the third generation of cephalosporin, is excreted into the biliary tract, and due to the drug interaction with calcium, it may lead to calcium-ceftriaxone precipitation [48]. Ceftriaxone-associated cholelithiasis has been observed with varying frequency (4.1–27.3%) in different studies [11,43,44,49,50]. The mean time of dissolution of stone in this group of patients was 9–12 months [44].

Another group at risk of cholelithiasis are patients taking long-term octreotide, a somatostatin analog, which may increase the proportion of biliary deoxycholic acid and inhibit gallbladder emptying. Gallstones or biliary deposits have been observed in nearly 33% of children with congenital hyperinsulinism during octreotide therapy. The occurrence of gallbladder pathology was independent of the dose and median age of initiation of octreotide therapy [51]. A retrospective analysis of childhood cancer survivors found that exposure to high-dose (>750 mg/m^2^) platinum chemotherapy, vinca alkaloid chemotherapy, or total body irradiation increased the risk of cholecystectomy in later life. Moreover, compared with healthy siblings, patients after cancer treatment had an approximately 30% higher risk of cholecystectomy [52]. In addition, prenatal exposure to toxins or drugs may affect gallbladder disorders in the offspring. Troisi et al. showed prenatal diethylstilbestrol (DES) exposure, a non-steroidal estrogen analog, was not associated with the risk of gallbladder disease overall or in sex-specific groups [53]. Contrary to this observation, Le et al. observed an elevated risk of gallstones among participants who were prenatally exposed to maternal smoking, especially among females and populations with low BMI [54].

### 2.5. Gender

Epidemiological studies have found that cholelithiasis is more common in adult women than men. A female predominance was also evident in most pediatric studies [9,43,44,50,55,56,57]. In only one study conducted in northern Iran, the majority of patients were boys (59.1%) [49]. Serdaroglu et al. evaluated the sex distribution by age groups and noted a higher incidence of gallstones in boys under 2 years of age and in girls over 10 years of age [44]. Tuna et al. observed a significantly higher age of girls than boys with cholelithiasis [50]. In both studies [44,50], no differences were found in terms of sex in the occurrence of symptoms. The impact of the higher incidence of cholelithiasis among girls may be related to puberty and the production of hormones, especially estrogen. Estrogens bound to estrogen receptors (ERs) in the liver and increase the secretion of cholesterol into the bile, promoting the formation of gallstones. Moreover, oral contraceptive use by girls may predispose to a more frequent occurrence of cholelithiasis.

### 2.6. Obesity

Obesity has been found to increase the risk of cholelithiasis development due to impaired gallbladder motility, excessive hepatic secretion, and bile saturation of cholesterol [58]. In the analysis of Frybova et al., among children hospitalized for laparoscopy, patients with cholelithiasis and choledocholithiasis, the mean BMI was significantly higher than in the control group without biliary stones on abdominal ultrasound [57]. Moreover, over a 9-year time period, the number of children with cholesterol cholelithiasis rose by 216%. Moreover, the mean BMI in children with cholesterol stones was higher than in children with hemolytic stones and biliary dyskinesia [9]. In the 20-year follow-up, there was an increase in the incidence of cholesterol gallstones in children from 27.3 to 70.6% and an increase in the mean BMI from 19.2 to 20.6 kg/m^2^ [56].

In cross-sectional study, a stronger association between obesity and gallstones was observed in girls than in boys [59]. Nunes et al. noticed intolerance to fatty foods in obese and overweight children with cholelithiasis. Children with abnormal BMI without cholelithiasis were less likely to complain of intolerance to dietary fats [60]. In another report of patients with symptomatic gallstone disease, an increase in obesity rate prevalence among children and adolescents, and there was an increase in hospitalization for cholelithiasis. Additionally, pediatric patients hospitalized with cholelithiasis were approximately six times more likely to be obese than children hospitalized with appendicitis [61].

Increased predisposition to the development of cholelithiasis in obese children may result from higher synthesis and excretion of cholesterol into the bile and impaired motility of the gallbladder [58,62]. Obesity in children also may predispose to the development of gallstone complications. A higher rate of pancreatitis in the pediatric population is associated with the more frequently observed obesity and cholelithiasis in children [63]. Based on the results of cross-sectional analysis, children with gallstone pancreatitis had statistically higher body weight percentile for age than patients with other causes of pancreatitis [64].

On the other hand, rapid weight loss (>1.5 kg/week) and a low-calorie diet may also predispose to gallstone formation due to the accelerated elimination of cholesterol, which over-saturates the bile [8]. In a study of obese children, 5.9% of children developed cholelithiasis after losing more than 10% of their body weight following a 6-month lifestyle change (physical activity, diet modification) [65]. In multivariate analysis, the decrease in BMI z-score and baseline BMI z-score correlated with the occurrence of gallstones. Interestingly, a rapid decrease in total cholesterol was associated with a higher risk of developing cholelithiasis. During an average of 4.8 years of follow-up, cholecystectomy was performed in 22% of children (two children with gallstones at baseline and three with gallstones developed during the program) with cholelithiasis, and no serious complications related to gallstone disease were observed [65]. Figure 1 presents the pathomechanism of the development of gallstone disease in obese pediatric patients and its influence on the course of cholelithiasis in obese children.

## 3. Conclusions

The increasing prevalence of cholelithiasis in children contributes to the growing interest in this disease. The incidence of gallstone disease in children is influenced by both genetic and environmental factors, such as exposure to certain drugs, e.g., ceftriaxone or octreotide. In addition, the presence of predisposing diseases can contribute to the formation of gallstones (such as primary sclerosing cholangitis [66], Wilson disease [67], renal stones [68], congenital nephrotic syndrome [69], hypothyroidism [70], Down syndrome [71,72], cystic fibrosis [73], Gaucher disease [74,75]). In recent years, there have been many studies on gallstone disease in children. There is still a lack of data on prophylaxis and treatment that would reduce the incidence of cholelithiasis in pediatric patients. It is important to take steps to reduce the incidence of obesity in children and adolescents. It is worth mentioning that the rapid reduction in BMI also promotes the formation of gallstones. Further studies are needed and may lead to the development of methods for the prevention of gallstones in pediatric patients.

## Figures and Tables

**Figure 1 ijms-23-13376-f001:**
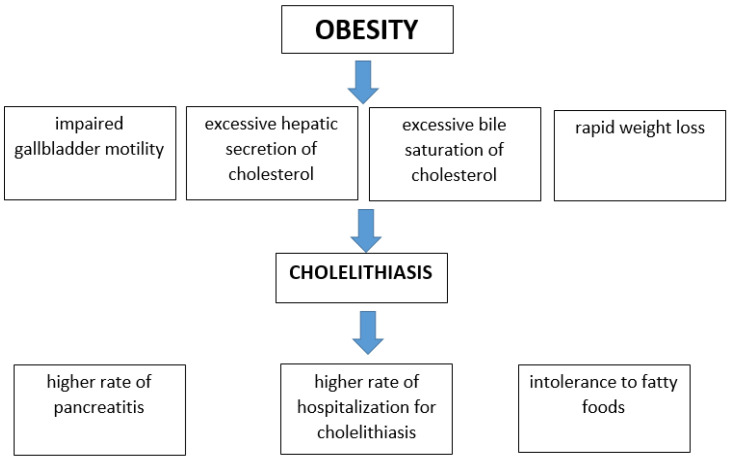
Influence of obesity on the development of gallstone disease and its complications in children.

**Table 1 ijms-23-13376-t001:** The genetic factors analyzed in the development of cholesterol gallstone disease were reported in children on the basis of the above-mentioned research.

Genetic Factor	Reference
*ABCG5*	[17,23,31]
*ABCG8*	[17,23,31]
*ABCG8* variant p.D19H	[17,22,24]
*NPC1L1* rs217434 polymorphism	[22]
*UGT1A1*	[22]
*ABCB4*	[26,31]
*ABCB4* variant c.2318G > T	[27]
*ABCB4* variant c.504C > T	[22]
*ABCB4* variant c.711A > T	[22]
*SLC10A1*	[28]
*SLC10A1* variant p.Ser267Phe	[29]
*ATP8B1*	[30]
*ABCB11*	[30]
*ABCC2*	[31]

## Data Availability

Not applicable.

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
