# Peer review of "The Etiology of Cholelithiasis in Children and Adolescents—A Literature Review"

_ijms, 2022, doi:10.3390/ijms232113376_

Round 1

Reviewer 1 Report (New Reviewer)

The abstract need rewriting as it is not clear.

The methodology of the identification of the literature/ methodological filter and no critique/ differentiation  of the literature reviewed.

It is presented logically with each possible etiology discussed.

This may be better titled etiology  of cholelithiasis in children and adolescents.

Author Response

Thank You for the evaluation of our manuscript. We are very grateful for Your valuable comments.

Please find our answers below.

  1. The abstract need rewriting as it is not clear.

Answer: Abstract has been changed.

The incidence of gallstone disease has increased in recent years. The pathogenesis of the cholelithiasis is not fully understood. The occurrence of the disease is influenced by both genetic and environmental factors. This article reviews the literature on cholelithiasis in children, with the exception of articles on hematological causes of cholelithiasis and cholelithiasis surgery. The aim of this review is to present the latest research on the pathogenesis of gallstone disease in children. The paper discusses the influence of all factors known so far, such as genetic predisposition, age, infections, medications used, parenteral nutrition and comorbidities on the development of gallstone disease. The course of cholelithiasis in the pediatric population is complex, ranging from asymptomatic to life-threatening. Understanding the course of the disease and predisposing factors can result in faster diagnosis of the disease and administration of appropriate treatment.

  1. The methodology of the identification of the literature/ methodological filter and no critique/ differentiation of the literature reviewed.

Answer: The methodology of the review has been added.

Line 105-112: . A review of MEDLINE/PubMed data was carried out in August 2022, using the phrase ‘cholelithiais’ or ‘gallstones’ in  combination with ’children’, ‘adolescents’ or ‘pediatrics’. Screening of the titles and abstracts was independently made by two investigators. The  studies included in this analysis fulfilled inclusion criteria: explored the etiology of cholelithiasis in pediatric patients and had available full texts. We excluded from the analysis articles on hematological causes of gallstone disease and surgical procedures for cholelithiasis. The selected papers were discussed with all authors.

  1. It is presented logically with each possible etiology discussed.

Answer: Thank you for your comment.

  1. This may be better titled etiology of cholelithiasis in children and adolescents.

Answer: The title has been changed.

The etiology of cholelithiasis in children and adolescents - a literature review.

Reviewer 2 Report (New Reviewer)

Why the majority of cases of fetal cholelithiasis were female? You said that the presence of gallstones in fetuses may be genetic causes of the disease. Could it be a syndrome present only in female?

In your article you did not describe the methodology of the review.

First you said that no studies describing the influence of diet on the occurrence of gallstone disease in children have been published so far, after that you have a section about obesity in children and how this increases the risk of cholelithiasis development.

In conclusions, also it is important to avoid the drugs which increase the risk of cholelithiasis (ceftriaxone)

Author Response

Thank You for the evaluation of our manuscript. We are very grateful for Your valuable comments.

Please find our answers below.

  1. Why the majority of cases of fetal cholelithiasis were female? You said that the presence of gallstones in fetuses may be genetic causes of the disease. Could it be a syndrome present only in female?

Answer: Currently, there are no data showing a direct influence of a genetic factor on the occurrence of gallstone disease in female fetuses. Due to the lack of published articles, this problem was not discussed in the article.

  1. In your article you did not describe the methodology of the review.

Answer: The methodology of the review has been added.

Line 105-112: . A review of MEDLINE/PubMed data was carried out in August 2022, using the phrase ‘cholelithiais’ or ‘gallstones’ in  combination with ’children’, ‘adolescents’ or ‘pediatrics’. Screening of the titles and abstracts was independently made by two investigators. The  studies included in this analysis fulfilled inclusion criteria: explored the etiology of cholelithiasis in pediatric patients and had available full texts. We excluded from the analysis articles on hematological causes of gallstone disease and surgical procedures for cholelithiasis. The selected papers were discussed with all authors.

  1. First you said that no studies describing the influence of diet on the occurrence of gallstone disease in children have been published so far, after that you have a section about obesity in children and how this increases the risk of cholelithiasis development.

Answer: Currently, no prospective studies evaluating the effect of diet on the development of cholelithiasis in children have been published. Indirectly, we can assume that the obesity epidemics are caused by diets high in sugar and saturated fat consumed in excessive amounts. However, there are no publications examining the direct impact of an unhealthy diet on the development of gallstone disease in children.

  1. In conclusions, also it is important to avoid the drugs which increase the risk of cholelithiasis (ceftriaxone)

Answer: The conclusion text has been supplemented with suggested information.

Line 330-332: The incidence of gallstone disease in children is influenced by both genetic and environmental factors like, such as exposure to certain drugs, e.g. ceftriaxone, or octreotide.

This manuscript is a resubmission of an earlier submission. The following is a list of the peer review reports and author responses from that submission.

Round 1

Reviewer 1 Report

none